# Alopecia Areata: A Review of the Role of Oxidative Stress, Possible Biomarkers, and Potential Novel Therapeutic Approaches

**DOI:** 10.3390/antiox12010135

**Published:** 2023-01-06

**Authors:** Lucia Peterle, Serena Sanfilippo, Francesco Borgia, Nicola Cicero, Sebastiano Gangemi

**Affiliations:** 1Department of Clinical and Experimental Medicine, Dermatology, University of Messina, Via Consolare Valeria-Gazzi, 98125 Messina, Italy; 2Department of Clinical and Experimental Medicine, School and Operative Unit of Allergy and Clinical Immunology, University of Messina, Via Consolare Valeria-Gazzi, 98125 Messina, Italy; 3Departement of Biomedical and Dental Sciences and Morphofunctional Imaging, University of Messina, 98168 Messina, Italy

**Keywords:** alopecia areata, oxidative stress, biomarkers, JAK inhibitors, nutraceuticals, autophagy

## Abstract

Alopecia areata (AA) is a dermatological condition characterized by non-scarring hair loss. Exact etiopathogenesis of AA is still unknown although it is known that several factors contribute to the collapse of the hair-follicle (HF)-immune-privileged (IP) site. Oxidative stress (OS) plays an important role in skin diseases. The aim of this review was to clarify the role of OS in AA pathogenesis and diagnosis, and to discuss potential treatment options. Oxidative-stress markers are altered in serum and skin samples of patients with AA, confirming a general pro-oxidative status in patients with AA. OS induces MHC class I chain-related A (MICA) expression in HF keratinocytes that activates the receptor NKG2D, expressed in NK cells and CD8+ T cytotoxic cells leading to destabilization of the HF immune-privileged site through the production of IFN-γ that stimulates JAK1 and JAK2 pathways. OS also activates the KEAP1-NRF2 pathway, an antioxidant system that contributes to skin homeostasis. In addition, a decrease of ATG5 and LC3B in the hair matrix and an increase in p62 levels indicates a reduction of intrafollicular autophagy during the evolution of AA. Potential biomarkers of OS in AA could be: malondialdehyde (MDA), advanced glycation end-products (AGEs), and ischemic-modified albumin (IMA). JAK inhibitors are the new frontier in treatment of AA and the use of nutraceuticals that modulate the OS balance, in combination with standard treatments, represent promising therapeutic tools.

## 1. Introduction

Alopecia areata (AA) is a common chronic, inflammatory, dermatological condition characterized by non-scarring hair loss. The worldwide prevalence of AA is approximatively 2% of the general population with no difference between different sexes, ages, or ethnicities. AA may affect people of any age but the first onset is most frequent in the third and fourth decades [1]. Most frequently patients present with smooth, circular, well-demarcated patches of complete hair loss that appear suddenly and occur over weeks. In some cases, hair loss can be extended to all the scalp (alopecia totalis (AT)) or all the body (alopecia universalis (AU)) [2]. Frequently the diagnosis of AA is clinically determined and, on dermoscopy, active disease is characterized by yellow dots, black dots, “exclamation marks”, or tapering hairs and broken hairs. In late or inactive stages of AA the presence of vellus hair is frequent [3]. AA can be associated with nail alterations such as trachyonychia and regular pitting, leukonychia, onycholysis, and onychomadesis [4]. In addition, eyes alterations such as Horner syndrome, pupil ectopia, iris atrophy, increased tortuosity of the fundus vessels, bilateral keratoconus, color changes in the iris, focal depigmentation in the iris, and choroid and retina pigment hyperplasia have been reported in AA [5]. AA is associated with several autoimmune diseases such as hyperthyroidism, hypothyroidism, goiter and thyroiditis, lupus erythematosus, vitiligo, psoriasis, rheumatoid arthritis, and inflammatory bowel disease [1]. The exact etiopathogenesis of AA is still unclear and several factors, such as genetic predisposition, autoimmunity, and environmental and emotional factors contribute to determining the disease [6,7,8]. The collapse of the hair follicle (HF)-immune-privileged (IP) site is a major precondition for the development of AA [9]. In detail, the loss of the HF-IP site is due to local increased interferon-γ (INF-γ) production, cytokines such as IL-15, IL-2, and CXCL2, upregulation of NKG2D ligands (e.g., MICA and ULBP3/6), and MHC I and MHC II molecules, in addition to the decrease of the IP guardians such as transforming growth factor-β1 (TGF-β1) and interleukin-10 (IL-10), [10,11,12]. All these factors contribute to the lost immune tolerance in the HF and consequential attack of autoantigens that are generated and expressed by melanocytes, keratinocytes, and/or dermal papilla fibroblasts during the anagen hair phase [11]. Specific autoantigens involved in the pathogenesis of AA are not yet defined but studies found that the synthetic epitopes derived from trichohyalin (structural protein) and tyrosinase-related protein-2 could be involved [13]. In addition, IFN-γ induces JAK/STAT signaling that can interfere with the hair growth cycle causing local inflammation resulting in a dystrophic anagen phase and a premature catagen phase [14].

It is already known that oxidative stress (OS) plays an important role in the pathogenesis of AA [15]. OS results from an imbalance between pro-oxidant and anti-oxidant mechanisms that coexist within cells to maintain system homeostasis. It occurs when the production of reactive species increases or when the activity of antioxidant is reduced. The reactive species include reactive oxygen species (ROS) and reactive nitrogen species (RNS). ROS include hydroxyl radical (OH-), hydrogen peroxide (H_2_O_2_), and superoxide anion (O2-). RNS include nitric oxide (NO-) and its derivative, peroxynitrite (ONOO−) which is produced by nitric oxide synthase (NOS). Both reactive oxygen and nitrogen species are called into action in the pathogenesis of AA. OS may be due to endogenous and exogenous stimuli (such as cigarette smoke, ultraviolet radiation, and chemical reactions) [16,17]. Other conditions that promote the production of reactive species include chronic inflammation, hypoxia, infections, physical and psychological stress, senescence, and trauma [18,19]. High levels of ROS cause release of proinflammatory cytokines, activation of transcription factors (MAPK, NF-kB, and AP1), apoptosis, and structural cellular alterations [20,21]. In small doses ROS are useful as a defense against infection but in large amounts they become harmful; for this reason, there are antioxidant enzymes that degrade them such as superoxide dismutase (SOD), catalase (CAT), glutathione peroxidase (GSH-Px), and paraoxonase (PON). OS produces alterations in lipids, proteins, and nucleic acids, inducing cellular damage [22]. Figure 1 provides a summary of the main markers related to oxidative-stress molecular targets [22]. Lipid peroxidation represents the hallmark of OS and malondialdehyde (MDA) and thiobarbituric acid reactive substances (TRABS) are the major markers of lipid peroxidation [15]. 

The aim of this paper was to review the studies available on the involvement of OS, in order to clarify its role in AA pathogenesis and diagnosis, and to discuss potential future treatment options.

## 2. Materials and Methods

The PubMed database (https://www.pubmed.gov, accessed on 13 December 2022) was used for this literature review. The search string used was: “alopecia areata” AND “oxidative stress”.

We then read the abstract of each article whose title suggested that the association between AA and OS was analyzed, until December 2022. The entire article was read only if the abstract indicated that the article potentially met our inclusion criteria: English language, research paper, human populations only, relevant to the outcome of interest, and/or full text available. Papers identified from the title, abstract, or full text as irrelevant to the topic in question were excluded. Finally, we examined the references of the articles selected in order to identify further studies that could be included in the review, based on the same criteria. After the searches, given the great variety of the topics dealt with and the results, it was not deemed appropriate to conduct a systematic review or a meta-analysis due to the low number of articles. Instead, we wrote this work in the form of a narrative review.

## 3. Results and Discussion

The preliminary PubMed search provided 41 articles for AA AND OS, 22 were not considered because they did not meet the inclusion criteria. Another 6 articles were added after the examination of the references of the articles selected.

For each of the 25 studies selected for inclusion in this review, Table 1 reports author(s) and year of publication, number of AA patients and controls, type of sample analyzed to detect OS markers (blood, skin biopsy, etc.), OS markers examined, and the main outcome(s) of each study.

### 3.1. OS Role in Pathogenesis of AA

To date, the etiopathogenesis of AA remains unclear, but it is generally accepted that it is an autoimmune disease. A sudden transition of the hair follicle from the anagen phase to the catagen phase is thought to underlie the clinical manifestation [48]. It is widely accepted that both genetic and environmental factors contribute to autoimmune conditions.

OS is known to be implicated in the pathogenesis of autoimmune disorders, including psoriasis (PS) [49], by causing inflammation, inducing cell apoptosis, and reducing immune tolerance. A recent systematic review and meta-analysis, in May 2022, demonstrated the existence of a bidirectional association between AA and PS. Both diseases are T cell-mediated (Th1/interferon, IL-23, and IL-17 were seen to be increased) [50]. OS is also involved in the pathogenesis of chronic inflammatory skin disorders, such as atopic dermatitis (AD). Chronic inflammation caused the rise of ROS and it also reduced the capacity of antioxidant systems (SOD, CAT, GSH, etc.), leading to alteration of DNA, lipids, and proteins resulting in production of MDA, NO, AOPPs, AGEs, etc. [51].

Several authors have investigated OS balance in AA.

Acharya et al., in 2019, conducted a systematic review and meta-analysis of 18 articles from the literature about OS in patients with AA; they observed that pro-oxidative indices (MDA, NO, and TOC) were increased, while antioxidant indices (SOD, PON, GSH-Px, and TAS) were reduced. In addition, they found a correlation between the increase of oxidation levels and the degree of disease severity [52].

Koca et al., in 2005, analyzed serum levels of MDA and NO and the serum activities of SOD and XO in patients with AA and control subjects. They found that the levels of MDA and NO and the activity of XO in AA serum patients were higher than in controls and that SOD activity was lower than in controls [24].

Bakry et al., for the first time, provided an evaluation of OS balance, measuring OSI and TOC in the blood of AA patients, confirming a general pro-oxidative status in AA patients [30]. Kim et al. measured ROM such as hydroperoxides in AA patients to test the free oxygen radicals and, by using the biological antioxidant potential (BAP) test, evaluated the plasma antioxidant potential. The mean levels of ROM in patients with AA were significantly higher in controls, and the mean level of antioxidant capacity in patients was significantly lower than in controls [25].

Cwynar, in 2019, investigated the role of OS in AA by measuring the levels of plasma and erythrocyte MDA and the CER in serum of 24 AA patients and comparing them with those of 24 healthy controls (HCs). They found significantly higher levels of MDA in AA patients. Malondialdehyde is a marker of oxidative stress in clinical situations, given that it informs about the degree of plasma-membrane damage as a result of the action of free radicals [36]. In support these results Bilgili et al., in 2013, in addition to serum TAC levels, TOS levels, and OSI evaluated the serum activity of PON1 an esterase capable of hydrolyzing lipid peroxide accumulation caused by OS. They found a decreased serum PON1 activity in AA patients suggesting that reduced PON1 activity may be related to an increase in oxidant and a decrease in antioxidant levels [28]. Similarly, Ramadan found a reduction of PON1 and Vitamin E levels in AA patients, confirming that an imbalance of the redox equilibrium takes part in the pathogenesis of AA [27].

In a recent study Nazli Dizen-Namdar compared TAS, TOS, and OSI levels, and PON1, ARE, and prolidase enzyme activity parameters of AA patients with results in line with those of previous studies [40]. Unlike previous studies, Akar et al., in 2001, evaluated levels of TBARS as lipid peroxidation status, and SOD and GSH-Px as antioxidant status in the scalp of 10 patients with AA. They found that the levels of TBARS, SOD, and GSH-Px in the scalp of patients with AA were significantly higher than those of controls. It is the author’s opinion that antioxidant molecules are high in AA patients most probably in defense against the excess production of superoxide radicals, especially in the early phase of disease. However, the TBARS levels were still high when compared with controls. These results suggest that the increased SOD and GSH-Px activities could not balance the elevated superoxide radical, or that the increase in TBARS levels might have originated from other radical(s) which are not metabolized by SOD and GSH-Px [23]. In contrast with these results, Alzolibani et al., in 2014, found that SOD activity in AA patient sera was significantly decreased compared to that in sera from healthy controls. These discrepancies may be attributed to differences among patients studied, such as disease duration and the pattern or severity of hair loss. Higher enzymatic antioxidant activity in patients with AA of shorter duration or lesser severity may be sufficient to act as a defense against excess production of ROS [31]. Interestingly, the same group of research, demonstrated that DNA modified by MDA was highly immunogenic in rabbits since perturbations in DNA by MDA generate neo-epitopes [53]. In the same year Rasheed et al. found that perturbations in SOD by NO generates neo-epitopes that might be one of the factors for the antigen-driven antibody induction in AA [32].

Although the majority of the studies found a disequilibrium in redox balance in AA patients, some authors did not find the same result: Motor et al., in 2018, found that TAS, TOS, and OSİ levels showed no significant difference between the control and AA groups suggesting that redox disequilibrium may take part in the pathogenesis of AA but several other physiopathological mechanisms are involved [29]. In addition, Khaki et al., in 2020, did not find a differences in serum level of GSH-Px and GSH-Rx between AA patients and controls [43]. Overall, most of the papers examined found an alteration of the redox balance in a pro-oxidative sense in patients with AA. The discrepancies in the results obtained could be related to the number of samples examined and the type of tissue examined.

The correlation between cytokine amounts and OS was provided in an interesting study produced by Tomaszewska et al. in 2020 [42]. In 33 Caucasian patients with AA and 30 healthy controls (HCs) IFN-γ, IL-1β, and IL-6 blood levels were measured. They found that the levels of these cytokines were significantly higher in AA patients than in HCs. TNF-α, IL-1β, and IFN-γ can induce nitric oxide synthase (iNOS), a cytotoxic effector molecule. Taskin S. et al., in 2022, compared the levels and activity of NO ·, ONOO^−^, and iNOS in 30 patients with AA and 30 healthy controls and found elevated values in patients with AA [47]. These results are in line with those previously found by Rasheed in 2014—they found that the average NO levels in AA patients’ sera were higher than in healthy controls [32]. These results confirm that patients with AA are exposed to potent nitrosative stress, and, in particular, to peroxynitrite, which acts as a bridge between ROS and RNS determining an amplification of the OS [47]. Table 2 summarizes the cited works, dividing them according to the finding or not of differences between the redox balance in AA patients and control groups.

Furthermore, IFN-γ is the key factor in OS mediated by xanthine oxidase- (XO-). Xanthine oxidase (XO) plays a key role in OS and its association with nitric oxide (NO)/NO synthase (NOS) has been reported [54]. IL-1β can also induce rapid expression of iNOS and generate large amounts of NO in tissues [55]. OS increases the expression of the intercellular adhesion molecule-1 (ICAM-1) in epithelial cells through the IL-6/AKT/STAT3/NF-κB-dependent pathway. Several studies indicate that upregulation of ICAM-1 expression in epithelial cells is closely associated with proinflammatory cytokines, such as IL-6, IL-1β, and tumor necrosis factor-α (TNF-α) [56]. All these facts suggest a link between OS and the serum cytokine profile in patients with AA. Interestingly the authors also compared these results with patients with non-segmental vitiligo with similar results supporting a common pathogenesis of both diseases [42].

Vitiligo, as AA, is a frequent immune-mediated disorder causing skin depigmentation, characterized by the selective loss of melanocytes which results in typical non-scaly, chalky-white macules. OS plays an important role in initiating vitiligo [57]. According to a recent review, OS contributes to melanocyte destruction causing DNA damage, lipid peroxidation, mitochondrial dysfunction, and endoplasmic-reticulum stress [58,59,60,61]. The increased levels of ROS, by altering the structure of autoantigens such as tyrosinase and MELAN-A, lead to the formation of new epitopes against which the immune response is triggered. OS plays a trigger role in initiating the pathophysiological process in vitiligo as is proved by an increase in oxidative-stress markers, such as lipid peroxidation, during the early stages of vitiligo and the increased levels of autoantibodies in the more advanced stages of the disease. In contrast, the specific antigens in AA have not yet been elucidated. However, since AA often affects pigmented hairs and regrowth is associated with depigmented hairs, it has been suggested that peptides associated with melanogenesis are involved in the AA [62].

Regarding the source of ROS in AA, several endogenous and exogenous factors can cause an accumulation of ROS in HF keratinocytes. As in vitiligo [63], in HF keratinocytes, OS probably induces MHC class I chain-related A (MICA) expression [64]. MICA is a distant homolog of major histocompatibility class I and it is the ligand for the activating receptor NKG2D, expressed in NK cells and CD8+ T cytotoxic cells [65]. In addition, the stressed environment downregulates IP guardians and the macrophage migration inhibitory factor (MIF) [66]. All these factors lead to destabilization of the HF immune-privileged site. CD8 + NKG2D + T-activated cells start to produce IFN-γ via JAK1 and JAK3 pathways. In follicular epithelial cells, IFN-γ stimulates the production of IL-15 activating JAK1 and JAK2. IL-15 then binds to the CD8+NKG2D+T cells to produce more IFN-γ, which amplifies the positive feedback loop, as shown in Figure 2 [67].

Another important pathway in oxidative stress homeostasis is the KEAP1-NRF2 axis.

This system codes for xenobiotic detoxification enzymes (phase II and phase III): physiologically in the cytoplasm, nuclear factor erythroid 2-related factor 2 (NRF2) is bound to kelch-like ECH-associated protein 1 (KEAP1). Oxidative stress causes conformational changes of KEAP1, whereby NRF2 eludes KEAP1 and translocates to the nucleus where it forms a heterodimer with sMAF binding to ARE with subsequent synthesis of phase II enzymes and phase III transporters.

In AD, this system also appears to be involved in maintaining skin homeostasis, regulating the state of desquamation and hyperkeratosis of keratinocytes [68].

There is a complex interaction between NRF2 and NF-κB, which helps to modulate several signal-transduction pathways that regulate oxidative state. NRF2 also appears to be involved in PS and vitiligo, modulating several cellular mechanisms (proliferation, keratinization, melanogenesis, and interaction between the skin and immune system) [69].

### 3.2. Autophagy Role in Pathogenesis of AA

Another mechanism potentially implicated in the pathogenesis of AA is the regulation of autophagy. Autophagy sets the skin cell population, stimulates anti-inflammatory activity, and plays an important role in cellular homeostasis.

Autophagy enables the destruction of organelles and macromolecules, recycling parts of them. It is ruled by several factors (oxidative stress, inflammation, hypoxia, etc.), trigged by external stimuli. LC3B is a trademark of the autophagosome autophagic membrane [70].

In particular, ROS can damage key proteins of the autophagy mechanism (ATG5 and Beclin) or it can alter signaling pathways such as JNK, p38, and Nrf2 [71].

Autophagy alterations are common to some immune-mediated skin diseases such as AA, PS, and AD. Alterations of autophagy-related proteins were observed in AA: analysis of some lesional AA samples showed a decrease in ATG5 and LC3B in the hair matrix and an increase in p62 levels. This indicates a reduction of intrafollicular autophagy during the evolution of AA.

The role of autophagy in other inflammatory skin diseases has been observed (e.g., PS and AD). Although the role of autophagy impairment is controversial, a reduction in it is known to occur in PS and AD [72].

A very recent review identified the melanocyte as the main culprit of autoimmune initiation in AA. Melanocyte metabolites can act as antigens, triggering the autoimmune attack. The authors also speculated that vitiligo and AA are two sides of the same coin (melanocytic impairment), resulting in alterations in autophagy mechanism. Furthermore, ROS production appears to underlie both diseases: ROS are produced in response to external stimuli but mainly during melanogenesis, so alterations in melanogenesis may underlie the pathogenesis of these diseases [73].

### 3.3. OS Biomarkers in AA

Given the potential role of OS in the pathogenesis of AA, the use of OS biomarkers could become not only a useful diagnostic tool, but also a guide for correct therapy and patient follow-up [74].

From our review of the current state-of-the-art data, potential biomarkers of OS to be considered in the AA could be:-malondialdehyde (MDA) [26,34,45];-advanced glycation end-products (AGEs) [46];-ischemic-modified albumin (IMA) [37,39];which are always found to be increased, by speculative analysis of currently existing studies. Other potential biomarkers evaluated were: superoxide dismutase (SOD), catalase (CAT), glutathione (GSH), and paraoxonase (PON).

In particular, MDA, the indicator of lipid peroxidation, is a very valuable biomarker and its quantification, with the most common analytical instruments, is easier to do than the other indicators of lipidic peroxidation [75]. In AA it could be useful as an indicator of the severity of the disease since high MDA levels were found in patients with clinically more severe forms of AA [26]. This was also confirmed in another study that tried to objectify disease severity using the SALT scoring. Results of this study demonstrated that with increasing disease severity as measured by SALT, serum MDA levels significantly increased, confirming a possible role of MDA as a parameter of disease severity or a marker for response to therapy [45]. High levels of MDA and 8-hydroxy-2′-deoxyguanosine (8-OHdG) have also been observed in AD, as markers of oxidative stress, associated with a reduction in PON [76].

Advanced glycation end-products (AGEs) are glycated proteins that induce the formation of ROS through the interaction with advanced glycation end-product-specific receptors and result in increased OS [77]. Shakoei et al., in 2022, evaluated the levels of AGEs in 40 patients with AA and 40 controls, they found that AGEs serum levels in AA patients were higher than in healthy controls. The role of AGEs in inflammatory diseases is also well studied in skin diseases [78,79]. In PS, an increase in AGEs and advanced oxidation protein-products (AOPPs) and a reduction in PON have been observed. In this instance, the transduction pathways involved were MAPK, NF-kB, AP-1, JAK-STAT, and other protein kinases [76]. In PS, the expression of the receptor for AGEs on epidermis, microvascular endothelium, inflammatory cells, and fibroblasts in the psoriatic plaques was higher than in perilesional and normal tissue [80]. The activation of these receptors leads to the expression of various transcription factors, especially NF-κB, that promote inflammatory processes in PS [81,82]. However, in PS, the levels of AGEs receptors do not correlate with the severity of the disease [80].

The majority of the reviewed studies envisaged the measurement of biomarkers at the serum level without comparing the values with the values acquired at the level of the tissues affected by the disease. Öztürk et al., in 2016, in line with previous studies, demonstrated significantly elevated activities of CAT, SOD, and adenosine deaminase (ADA), as well as increased GSH and MDA in the lesions of patients with AA compared to the control group. A peculiarity of this study, as with the study by Akar et al. [23], is that the levels of these molecules were measured in the skin samples affected by the disease. Although a concordance between tissue and serum levels suggests a possible use of these molecules as a marker of disease, further studies are necessary for confirming this correlation.

In AD, the STAT3/NF-kB pathway is the target of ROS. High levels of MDA and 8-OHdG have also been observed in AD as markers of OS, associated with a reduction in PON [76].

Finally, assaying the activity of SOD, CAT, GSH, and PON could also be useful, but the values of these enzymes could be affected by the time of onset of AA, thus being confounding at the moment.

IMA seems to be a marker of OS in several dermatological diseases, as shown by a recent review [20]. Ataş et al. demonstrated that IMA is an independent predictor of OS in patients with AA [37]. In addition, they found that the sensitivity, specificity, and positive and negative predictive values and capacity of IMA were higher than the values of SOD, CAT, glutathione-S-transferase (GSH-ST), and MDA [37].

### 3.4. Potential Therapeutic Target in AA

Given that OS appears to contribute to the pathogenesis of alopecia areata, the use of molecules that influence the redox balance could represent not only a useful tool in the prevention of possible relapses but also a useful tool to support therapies aimed at the molecular targets at the basis of the onset of AA. The main culprit cytokines in AA seem to be IFN-γ and IL-15, activated by the JAK-STAT transcription pathways.

IL-15 binds both to IL-15Rα (expressed on follicular epithelium cells) and to IL-15Rβ (expressed on NKG2D + CD8+ T cells). IL-15 activates downstream pathways such as JAK1/3-STAT5, MAPK, and mTOR, having cytotoxic action. Activated NKG2D+CD8+ T cells produce IFN-γ in follicular epithelium cells; IFN-γ increase the expression of MHC I and II and it induces the secretion of CXCL9, CXCL10, and CXCL11 (CXCR3 ligands), through the JAK1/2-STAT1 pathway [73].

JAK/STAT pathways can be inhibited by vitamin D supplementation. Vitamin D also increases the differentiation of Tregs cells by inducing the expression of CD25 and Foxp3, promoting the immune tolerance mechanism in HF. Vitamin D use in some clinical trials showed an increase in TAC and GSH, and a reduction in MDA. Vitamin D acts on NRF2 transcription through VDR, it maintains redox homeostasis and it is primarily responsible for hair growth. Vitamin D also reduces the expression of CXCL-10 and TLRs, downregulating the T-cell-mediated immune response [15]. As regards NRF2, it has been demonstrated that its activation can contribute to the reduction of ROS-associated damage through upregulation of proteins involved in the direct clearance of H2O2 [83]. In addition to Vitamin D, numerous NRF2-targeting agents have been identified, most notably sulforaphane (SFN) but also other phytochemicals such as curcumin, silymarin, and resveratrol, which can serve as candidates for upregulation of intrafollicular NRF2 activity [84]. Therefore, further studies should be carried out with the aim of investigating the use of these molecules for both topical and systemic use.

Additionally, since both AD and PS share defects in autophagy, the use of JAK/STAT inhibitors was introduced, achieving good results. Therefore, the use of such drugs could be suggested for AA as well [72].

Several studies have shown the potent effect of JAK/STAT inhibitors. JAKs are IFN-γ-receptors downstream effectors; these receptors are expressed in both capillary follicle and immune cells and their downstream inhibition promotes hair stem-cell proliferation and angiogenesis. Ruxolitinib (JAK1 and JAK2 inhibitor) and tofacitinib (JAK1 and JAK3 inhibitor) are the most used: the first blocked IL-17 axis and the second reduced CXCL10 levels leading to decrease of CD8+ T cell, induced angiogenesis and VEGF production [85].

A recent systematic review and meta-analysis shows the efficacy of oral use of tofacitinib and ruxolitinib in adult patients with AA, especially in the long term. Regarding possible side effects, longer treatment durations of over six months were associated with an increased risk of laboratory abnormalities such as alterations in hemoglobin, blood cell count, liver transaminase, and lipids. The other most common adverse events (infection, neurological symptoms, gastrointestinal symptoms, and cutaneous symptoms) are not associated with treatment duration. However, discontinuation of the drug resulted in a recurrence of hair loss within three months of interruption. Therefore, maintenance therapy with lower doses may be explored to prevent recurrence and more studies on long-term side effects are needed [86]. The use of antioxidant molecules in the prevention of possible relapses should be further investigated. This could allow a reduction in the dosage of JAK inhibitors and therefore their more prolonged use or even their suspension.

There is also an efficacy and safety study on the use of tofacitinib, even in the pediatric population, that have demonstrated the usefulness of this drug in patients with refractory AA [87].

Finally, given that it has been shown that there is an alteration of the redox balance in subjects affected by AA, the association of antioxidant molecules with the currently used therapies could be useful.

Abbas treated 20 AA patients with ginger powder, with an improvement in the antioxidant status of the erythrocytes and lymphocytes and an increase in the serum zinc levels [41]. Several studies demonstrated that zinc levels are low in AA patients [88,89,90], therefore studies demonstrating its usefulness as a supplement in patients with AA are needed.

A 2020 pilot study, in which healthy HF in anagen scalps were treated ex vivo with low doses of IFN-γ, showed an increase in MHC class I, a decrease in ATG5 and LC3B (markers of autophagy), and an increase in p62. Subsequently, the same sample was treated with methyl-spermidine, detecting an increase in anagen phase and stimulation of autophagy [91].

In a 2011 murine study, mice with spontaneous AA were treated with subcutaneous quercetin injections that showed hair regrowth in the lesional area, comparing them with mice treated with sham injections. A decrease in HSP70 was also observed, so it can be hypothesized that quercetin blocks HSP70 induction, reducing the production of pro-inflammatory cytokines through NF-kB activation. Quercetin is a bioflavonoid with anti-inflammatory properties [92].

A 2018 review of literature about micronutrients showed that vitamin D, zinc, and folate levels are reduced in AA patients, at the moment, we have inconsistent data on iron, copper, selenium, magnesium, and vitamin b12 levels. Clinical trials should be conducted in the future to understand the role of micronutrients in AA [93].

## 4. Conclusions

OS probably plays a relevant role in demonstrating the imbalance of redox state in patients with AA. The exact mechanism by which this happens remains to be defined and further studies with larger groups are required. The prevalent hypothesis is the role of melanocytes in the loss of immune privilege and the production of ROS during the process of melanogenesis. More studies need to be conducted to monitor changes in oxidative-stress markers before and after therapeutic treatment. The use of new therapeutic approaches beyond traditional ones is desirable. JAK inhibitors are the new frontier in consideration of the fact that they may be useful in treating multiple autoimmune diseases together.

## Figures and Tables

**Figure 1 antioxidants-12-00135-f001:**
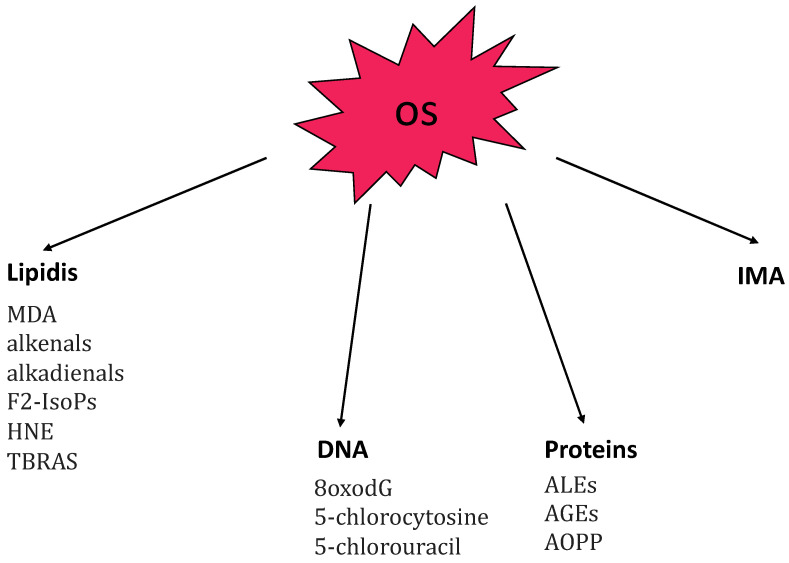
Summary of the main markers related to oxidative stress (OS) principal molecular targets: lipidis, DNA, proteins, and IMA. Abbreviations: 8oxodG, 7,8-dihydroxy-8-oxo-2′-deoxyguanosine; AGEs, advanced glycation end-products; ALEs, advanced lipoxigenation end-products; AOPP advanced oxidation protein-products; F2-IsoPs, F2-isoprostanes; HNE, 4-hydroxy-2-nonenal; IMA, ischemia-modified albumin; MDA, malondialdehyde; TBARS, thiobarbituric acid reactive substances.

**Figure 2 antioxidants-12-00135-f002:**
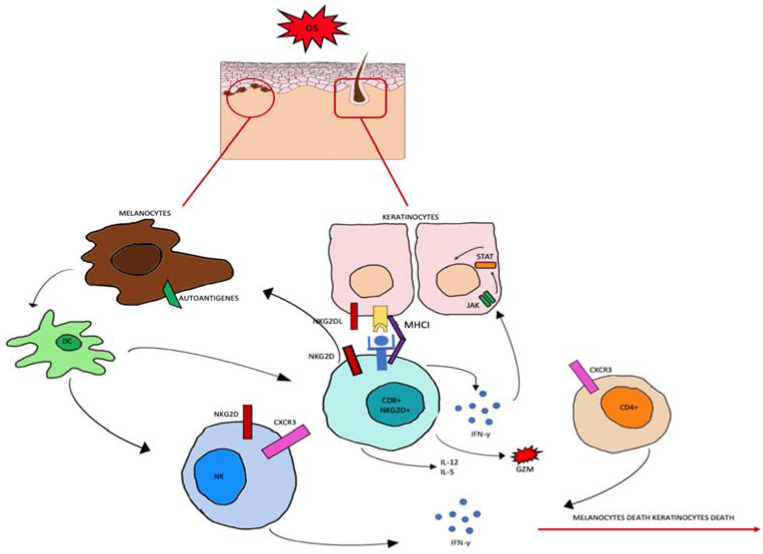
Oxidative stress (OS) induces the production of autoantigens and the expression of NKG2DL and MHCI in keratinocytes and melanocytes. CD8+ cells expressing NKG2D become effector cells by binding to the NKG2D receptor. CD8+ cells produce IL-2 and IL-15, granzyme b (GZM), and interferon gamma (INF-γ). INF-γ activates the JAK-STAT signaling pathway which can induce further abnormal expression.

**Table 1 antioxidants-12-00135-t001:** Summary of the articles concerning AA and OS in chronological order, from oldest to most recent. For each study, in addition to the author and year of publication, the number of patients, tissues, markers of oxidative stress, and oxidative stress main outcome(s) are also reported.

Authors (Reference)	Year	Number of Patients	Tissues	Markers of Oxidative/Stress	Oxidative Stress Main Outcome(s)
Akar A. [23]	2002	10 AA 10 HC	Scalp samples	TBARS, SOD, GSH-Px	TBARS, SOD, and GSH-Px were significantly higher than those of controls levels of TBARS, SOD, and GSH-Px in early phase of disease
Koka [24]	2005	24 AA 20HC	Blood samples	MDA, NO (nitrite/nitrate), XO and SOD activity	MDA, NO (nitrite/nitrate), and XO activity levels were higher in AA patients than in controls; SOD activity was lower
Kim S. W. [25]	2010	16 pz AA_ 16 HC_	Blood samples_	ROM, AC	ROM were increased and AC was reduced versus controls
Fattah N. S. A. [26]	2011	50 AA 50 HC	Punch biopsies	MDA, SOD	MDA levels were high in AA patients and SOD activity was low
Ramadan S. [27]	2012	15 AA 15 HC	Punch biopsies Blood samples	PON1, Vitamin E	Lower tissue and serum PON1 and Vitamin E levels in the patients than in the controls
Bilgili S. G. [28]	2013	39 AA 39 HC	Blood samples	PON1, TAC, TOS, OSI	TAC levels and PON1 activity were lower in AA patients than in controls; TOS levels and OSI were significantly higher
Sedat Motor [29]	2014	46 AA 36 hc	Blood samples	TAS, TOS	TAS, TOS, and OSİ levels showed no significant difference between the control and AA groups
Bakry O. A. [30]	2014	35 AA 30 HC	Blood samples	OSI, TOC, TAC, MDA	TOC, MDA, and OSI were high in AA patients and TAC value was low; higher MDA and OSI values and lower TAC values were found in severe AA than in mild or moderate AA
Alzolibani A. A. [31]	2014	26 AA_ 30 HC_	Blood samples	SOD	SOD activity was reduced in AA patients compared to controls
Rasheed Z. [32]	2014	26 AA_ 30 HC_	Blood samples	NO, SOD	NO was elevated and SOD activity was reduced in AA patients compared to controls
Kalkan G. [33]	2015	119 AA 104 HC		nSOD Ala-9Val and GPx1 Pro 198 Leu polymorphisms and AA susceptibility	nSOD Ala-9Val SNP genotype distributions and allele frequencies of the AA patients and the control group
Yenin J. Z. [34]	2015	62 AA 62 HC	Blood samples	MD and CAT, SOD, GSH-Px	No statistically significant difference in patient plasma MDA levels, CAT, GSH-Px, or SOD activities with regard to AA severity, duration, recurrence, or pattern
Perihan Öztürk [35]	2016	30 AA 30 HC	Scalp-scrapes	ADA	Factors associated with oxidative stress were elevated in AA patients
Cwynar A. [36]	2018	24 AA 22 HC	Blood samples	MDA, CER	MDA high in AA patients
Ataş H. [37]	2019		Blood samples	IMA	IMA levels increased in AA patients
Cwynar A. [38]	2019	30 AA 30 HC	Blood samples	PON1, MDA, AOPPs	
Incel-Uysal P. [39]	2019	35 AA 35 HC	Blood samples	IMA, sd-LDL, and visfatin levels	IMA levels increased in AA patients
Nazli Dizen-Namdar [40]	2019	60 AA, 50 HC	Blood samples	Serum PON1, prolidase, arylesterase activities, TOS, TAS, OSI	TOS and OSI levels and prolidase were high in AA patients; PON1 and arylesterase activities were low; no difference in serum TAS levels between the two groups
Abbas A. N. [41]	2020	20 AA	Blood samples	GSH, MDA, TAS	Improvement of the antioxidant/oxidant balance of the erythrocytes and lymphocytes
Tomaszewska K. [42]	2020	30 AA 30 Vit 30 HC	Blood samples	IFN-*γ*, IL-1*β*, IL-6	Oxidative stress may play a significant role in promoting and amplifying the inflammatory process both in AA and vitiligo
Khaki L. [43]	2020	56 AA 19 HC	Blood samples	GSH-Px, GSH-Rx	No differences in serum levels of glutathione reductase and glutathione peroxidase between the two groups
Mustafa A. I. [44]	2021	49 AA 49 HC	Serum	8-OHdG, HMBG1, CRP	High levels correlated with disease gravity
Sachdeva S. [45]	2022	40 AA, 40 HC	Blood samples	MDA, SOD, TAS	TAS and SOD were lower in AA patients, MDA was higher
Shakoei S. [46]	2022	40 AA, 40 HC	Blood samples	Blood sugar, C-reactive protein, lipid profile, and AOPPs, AGEs, PON1, lecithin-cholesterol acyltransferase and serum ferric-reducing antioxidant power	Advanced glycation end-products and advanced oxidation protein-products were significantly higher in patients with alopecia areata
Taskin S. [47]	2022	30 AA 10 HC	Blood samples	NO, ONOO^−^, NOS activity	NO, ONOO^−^, and NOS activity were significantly higher in AA patients with than in the control group

Abbreviations: GSH, glutathione; MDA, malondialdehyde; TAS, total antioxidant status; TOS, total oxidative status; OSI, oxidative stress index; SOD, superoxide dismutase; CER, ceruloplasmin; ADA, adenosine deaminase; IMA, ischemia-modified albumin; CAT, catalase; GSH-Px, glutathione peroxidase; GSH-Rx, GSH reductase; AOPPs, advanced oxidation protein-products; NO, nitric oxide; ONOO^−^, peroxynitrite; NOS, nitric oxide synthase; TBARS, lipid peroxidation status; XO, xanthine oxidase; ROM, reactive oxygen metabolites; AC, antioxidant capacity; AGEs, advanced glycation end-products; AOPPs, advanced oxidation protein-products.

**Table 2 antioxidants-12-00135-t002:** Summary of the works dividing them into those in which a difference in the redox balance between the AA patients and the controls was found and those in which no difference was found. The majority of the studies found an imbalance of redox equilibrium.

Difference in Redox Balance between AA Patients and Healthy Controls		No Difference in Redox Balance between AA Patients and Healthy Controls	
Authors	Markers of OS	Authors	Markers of OS
Bakry et al., 2014	OSI, TOC	Motor et al., 2014	TAS, TOS, OSI
Cwynar et al., 2018	MDA, CER	Khaki et al., 2020	GSH-Px, GSH-Rx
Bilgili et al., 2013	TAC, TOS, OSI, PON1		
Nazli Dizen-Namdar et al., 2019	TAS, TOS, OSI, PON1, ARE, prolidase enzyme activity		
Akar et al., 2001	TBRAS, SOD, GSH-Px		
Tomaszewska. et al., 2020	iNOS		
Taskin et al., 2022	NO, ONOO^−^, iNOS		
Rasheed. et al., 2014	NO, SOD		
Kim et al., 2010	ROM, AC		
Koca et al., 2005	MDA, NO, SOD, XO		

Abbreviations: MDA, malondialdehyde; TAS, total antioxidant status; TOS, total oxidative status; OSI, oxidative stress index; TAC, total antioxidant capacity; TOC, total oxidant capacity; SOD, superoxide dismutase; CER, ceruloplasmin; GSH-Px, glutathione peroxidase; NO, nitric oxide; ONOO^−^, peroxynitrite; iNOS, inducible nitric oxide synthase; TBRAS, lipid peroxidation status; PON1, paraoxonase 1; ROM, reactive oxygen metabolites; AC, antioxidant capacity; XO, xanthine oxidase.

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
