# Peer review of "Alopecia Areata: A Review of the Role of Oxidative Stress, Possible Biomarkers, and Potential Novel Therapeutic Approaches"

_antioxidants, 2023, doi:10.3390/antiox12010135_

Round 1
Reviewer 1 Report
In Ms-antioxidant-2129818 the authors analyze the results of 17 selected studies, producing a detailed analysis of the aspects relating oxidative stress to alopecia areata, the study can be accepted with minor revisions:
- Table 1 is not readable, it is necessary to modify it by increasing the resolution of the image or, alternatively, redo it.
- The statements reported in the introductory part "OS may be due to endogenous and exogenous stimuli (such as cigarette smoke, ultraviolet radiations, chemical reactions", must be completed with bibliographic references.
- On Page 5, the list of the cited works could be collected in the table as done for table 1, thus collecting the information in a summary way.
- Regarding the potential therapeutic targets in the treatment of AA, are any side effects known in the long-term use of the molecules mentioned: Tofaticinib and Auxolitinib?
- The text must be meticulously double-checked for the presence of errors of form and typos.
Author Response
Dear Reviewer,
Thank you very much for Your consideration and the suggestions.
Appropriate changes were made and highlighted in the revised manuscript according to the observations. In the following, we inserted our point-by-point responses to your comments:
- Table 1 is not readable, it is necessary to modify it by increasing the resolution of the image or, alternatively, redo it.
Table 1 has been inserted in a more readable format.
- The statements reported in the introductory part "OS may be due to endogenous and exogenous stimuli (such as cigarette smoke, ultraviolet radiations, chemical reactions", must be completed with bibliographic references.
Bibliographic references were added in the introductory part as suggested.
- On Page 5, the list of the cited works could be collected in the table as done for table 1, thus collecting the information in a summary way.
A new table (Table 2) was edited to collect the information in a summary way.
- Regarding the potential therapeutic targets in the treatment of AA, are any side effects known in the long-term use of the molecules mentioned: Tofaticinib and Ruxolitinib?
Longer treatment durations of over six months are associated with an increased risk of laboratory abnormalities such as alterations in hemoglobin, blood cell count, liver transaminase and lipids. The other most common adverse events (infection, neurological symptoms, gastrointestinal symptoms and cutaneous symptoms) are not associated with treatment duration. We have added this data in the text. Interestingly, a relapse was noted after the suspension of the treatment, this suggests that the therapy with JAK inhibitors should be prolonged for long periods or supported with other molecules.
- The text must be meticulously double-checked for the presence of errors of form and typos.
The text was checked for typos and errors of form.
Reviewer 2 Report
The authors have performed a review on the role of oxidative stress in the pathogenesis of alopecia areata. However the topic is not novel (DOI: 10.1111/ijd.14753). Regarding this review there are several major issues.
In the Introduction Section, the authors mention "OS produces alterations of lipids, proteins and nucleic acids, inducing cellular damage". In the next sentence, the authors mention only lipid peroxidation markers. The authors should summarize the main markers related to oxidative stress molecular targets. A figure including these markers might be useful.
"The exact etiopathogenesis of AA is still unclear and several factors contribute to determine the disease such as genetic predisposition, autoimmunity, environmental and emotional factors." The authors should provide the bibliographic source.
The authors should specify why they used PubMed as the only database and define the type of review they performed.
The inclusion and exclusion criteria are mentioned both in the Material and Methods Section and in the Results Section. However, the criteria are different between the two sections, more criteria being mentioned in the Results Section. (e.g. the studies that were not performed on humans were excluded). The authors should define more clearly the inclusion and exclusion criteria of the studies analyzed and list them only in the Material and Methods Section.
The authors have mentioned that they included 17 studies in the review, but there are only 14 studies listed in Table 1. The table is difficult to read, a higher resolution is needed. The article by Kalkan et al. listed in Table 1 is not found in the list of bibliographic references. This article seems to have the same bibliographic index as the article by Mustafa et al. (13).
Table 1 should be revised. For example the article by Mustafa et al. was published in 2021, not in 2012.
The title of Table 1 should be revised. The title should reflect the content of the table, but not include the names of table columns.
Other articles evaluating oxidative stress in alopecia areata are available in PubMed, but are not included in this review (e.g. PMID: 15917721). Why were they excluded?
The first paragraph of Section 3.1.has no bibliographic sources. There are several such paragraphs without a bibliographic source. The authors should revise the entire manuscript.
The authors should detail the link between autophagy and oxidative stress. Otherwise, it is not clear to the reader why the Autophagy role in pathogenesis of AA Section was included.
I don't understand the 3.3) OS biomarkers in the AA Subsection. Only three oxidative stress markers (MDA, AGE, IMA) are mentioned. It is not clear why. Some of the studies that are cited are not found in Table 1 and some are not related to alopecia areata (e.g. 51, 52, etc).
Section 3.4) Potential therapeutic target in AA. The connection between oxidative stress and this section is unclear. The authors should provide more data.
The first sentence in the Conclusions Section should be removed.
Author Response
Dear Reviewer,
Thank you very much for Your consideration and the suggestions.
Appropriate changes were made and highlighted in the revised manuscript according to the observations. In the following, we inserted our point-by-point responses to your comments:
- In the Introduction Section, the authors mention "OS produces alterations of lipids, proteins and nucleic acids, inducing cellular damage". In the next sentence, the authors mention only lipid peroxidation markers. The authors should summarize the main markers related to oxidative stress molecular targets. A figure including these markers might be useful.
A figure including markers related to oxidative stress molecular target was added as suggests.
- "The exact etiopathogenesis of AA is still unclear and several factors contribute to determine the disease such as genetic predisposition, autoimmunity, environmental and emotional factors." The authors should provide the bibliographic source.
Bibliographic source was added at the sentence.
- The authors should specify why they used PubMed as the only database and define the type of review they performed.
Given the great variety of the topics dealt with and the low number of results, it was deemed appropriate to conduct a narrative review as we added to the text.
- The inclusion and exclusion criteria are mentioned both in the Material and Methods Section and in the Results Section. However, the criteria are different between the two sections, more criteria being mentioned in the Results Section. (e.g. the studies that were not performed on humans were excluded). The authors should define more clearly the inclusion and exclusion criteria of the studies analyzed and list them only in the Material and Methods Section.
Inclusion and exclusion criteria of the studies analyzed were list more clearly only in Material and Method section.
- The authors have mentioned that they included 17 studies in the review, but there are only 14 studies listed in Table 1. The table is difficult to read, a higher resolution is needed. The article by Kalkan et al. listed in Table 1 is not found in the list of bibliographic references. This article seems to have the same bibliographic index as the article by Mustafa et al. (13).
Table 1 has been inserted in another format, a correct reference for the article by Kalkan was added and typing errors revised.
- Table 1 should be revised. For example, the article by Mustafa et al. was published in 2021, not in 2012.
Table 1 was revised.
- The title of Table 1 should be revised. The title should reflect the content of the table, but not include the names of table columns.
The title of Table 1 was revised.
- Other articles evaluating oxidative stress in alopecia areata are available in PubMed, but are not included in this review (e.g. PMID: 15917721). Why were they excluded?
The string used for our search "Alopecia Areata AND Oxidative Stress” produced a total of 41 results, in which, some articles such as the one mentioned are not present.
We then carefully examined the references of the articles selected from the first search, thus adding more articles to the results. These articles have been reported in Table 1 and the research method has been better specified in the Materials and Methods Section.
- The first paragraph of Section 3.1.has no bibliographic sources. There are several such paragraphs without a bibliographic source. The authors should revise the entire manuscript.
Bibliographic source for the first paragraph of Section 3.1. was added and entire manuscript was revised.
- The authors should detail the link between autophagy and oxidative stress. Otherwise, it is not clear to the reader why the Autophagy role in pathogenesis of AA Section was included.
The alteration of the mechanisms involved in autophagy is a relevant aspect in AA pathogenesis of and a current object of study. There is a tight interaction between OS and autophagy, for this reason we wanted to highlight this topic by dedicating a separate section to it. More details about the link between autophagy and OS were added as suggest.
- I don't understand the 3.3) OS biomarkers in the AA Subsection. Only three oxidative stress markers (MDA, AGE, IMA) are mentioned. It is not clear why. Some of the studies that are cited are not found in Table 1 and some are not related to alopecia areata (e.g. 51, 52, etc).
In the subsection 3.3, we discuss the potentially more useful and usable OS biomarkers in AA, based on what emerged from the literature review and also comparing them with those used in other skin diseases such as, for example, psoriasis. At the end of the subsection, we also discuss the role of SOD, CAT, GSH and PON. Typos concerning citations have been revised.
- Section 3.4) Potential therapeutic target in AA. The connection between oxidative stress and this section is unclear. The authors should provide more data.
More data have been added to the section 3.4 as suggest.
- The first sentence in the Conclusions Section should be removed.
The sentence was removed.
Round 2
Reviewer 2 Report
All suggested changes have been taken into consideration by the authors; the manuscript has been significantly improved.